# Quantifying Inhaled Concentrations of Particulate Matter, Carbon Dioxide, Nitrogen Dioxide, and Nitric Oxide Using Observed Biometric Responses with Machine Learning

Shisir Ruwali, Shawhin Talebi, Ashen Fernando, Lakitha O. H. Wijeratne, John Waczak, Prabuddha M. H. Dewage, David J. Lary *, John Sadler, Tatiana Lary, Matthew Lary and Adam Aker

Department of Physics, The University of Texas at Dallas, Richardson, TX 75080, USA;
shisir.ruwali@utdallas.edu (S.R.); lhw150030@utdallas.edu (L.O.H.W.); john.waczak@utdallas.edu (J.W.);
pxh180012@utdallas.edu (P.M.H.D.); tlary@me.com (T.L.); matthew.lary@utdallas.edu (M.L.)
* Correspondence: david.lary@utdallas.edu

**Abstract:** Introduction: Air pollution has numerous impacts on human health on a variety of time scales. Pollutants such as particulate matter—$PM_1$ and $PM_{2.5}$, carbon dioxide ($CO_2$), nitrogen dioxide ($NO_2$), and nitric oxide (NO) are exemplars of the wider human exposome. In this study, we adopted a unique approach by utilizing the responses of human autonomic systems to gauge the abundance of pollutants in inhaled air. Objective: To investigate how the human body autonomically responds to inhaled pollutants in microenvironments, including $PM_1$, $PM_{2.5}$, $CO_2$, $NO_2$, and NO, on small temporal and spatial scales by making use of biometric observations of the human autonomic response. To test the accuracy in predicting the concentrations of these pollutants using biological measurements of the participants. Methodology: Two experimental approaches having a similar methodology that employs a biometric suite to capture the physiological responses of cyclists were compared, and multiple sensors were used to measure the pollutants in the air surrounding them. Machine learning algorithms were used to estimate the levels of these pollutants and decipher the body's automatic reactions to them. Results: We observed high precision in predicting $PM_1$, $PM_{2.5}$, and $CO_2$ using a limited set of biometrics measured from the participants, as indicated with the coefficient of determination ($R^2$) between the estimated and true values of these pollutants of 0.99, 0.96, and 0.98, respectively. Although the predictions for $NO_2$ and NO were reliable at lower concentrations, which was observed qualitatively, the precision varied throughout the data range. Skin temperature, heart rate, and respiration rate were the common physiological responses that were the most influential in predicting the concentration of these pollutants. Conclusion: Biometric measurements can be used to estimate air quality components such as $PM_1$, $PM_{2.5}$, and $CO_2$ with high degrees of accuracy and can also be used to decipher the effect of these pollutants on the human body using machine learning techniques. The results for $NO_2$ and NO suggest a requirement to improve our models with more comprehensive data collection or advanced machine learning techniques to improve the results for these two pollutants.

**Keywords:** autonomic response; exposome; microenvironment; air pollution; biometric observations; machine learning; particulate matter; $CO_2$; $NO_2$; NO

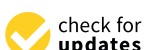



## 1. Introduction

This study employed a novel approach to gauge the levels of pollutants found in inhaled air using autonomic human responses as discerned using a suite of biometric sensors. The environmental and social context has a significant impact on human well-being. The issue of air pollution is of particular concern, as reported by the World Health Organization's findings that both outdoor and indoor pollution contribute to more than 7 million premature deaths each year [1]. Air pollution can come from various sources, including natural events such as wildfires and volcanic eruptions, as well as human

activities such as vehicle emissions, industrial processes, and the operation of coal-fueled power plants.

The air quality standards established by the U.S. Environmental Protection Agency under the Clean Air Act include six pollutants. These include particulate matter (PM), carbon monoxide (CO), ground-level ozone, nitrogen dioxide ($NO_2$), sulfur dioxide ($SO_2$), and lead [2]. Some of the other pollutants include carbon dioxide ($CO_2$) and volatile organic compounds. Particulate matter refers to minuscule solid or liquid particles that are present in the air and are categorized on the basis of their aerodynamic diameter. They include $PM_{1.0}$, $PM_{2.5}$, and $PM_{10}$ with an aerodynamic diameter of less than 1 μm, 2.5 μm, and 10 μm, respectively. With the small size of $PM_{2.5}$, these particulates can penetrate deeply into the lungs and bloodstream, creating adverse health effects related to the respiratory system [3], increased mortality [4], heart disease [5], inflammatory responses, and adverse birth-related effects [6]. The pollutants we considered are exemplars of the wider human exposome [7–9], which refers to the comprehensive accumulation of all environmental exposures that an individual encounters throughout their lifetime, including chemicals and biological agents. The exposome encompasses exposures to both gases and particulates, and appropriate care should be taken to include the often ignored ultrafine particulates [10].

Guidelines on the recommended levels of exposure to pollutants provided by the World Health Organization (WHO) [11] and the Environmental Protection Agency (EPA) [12] contain only two designations: short-term exposures (an average of over 24 h) and long-term exposures (a 1-year average). Brief daily encounters, such as passing a construction site, walking on a busy road, or even working in poorly ventilated indoor spaces, can expose individuals to levels higher than the recommended guidelines. The size of airborne PM has a major influence on how far it can penetrate the lungs, which in turn affects human health. The WHO acknowledges that PM with diameters below 2.5 μm ($PM_{2.5}$) has a significant disease burden on human health [11,13], while larger particles, although less likely to reach the alveoli, can still cause health problems by irritating the eyes, nose, and throat [11]. Therefore, research efforts focused on prolonged exposure to poor air quality, including airborne particles of varying sizes, are of particular importance when considering long-term health.

The area of respiratory health receives significant attention due to the high incidence of poor air quality caused by factors such as smoke, vehicle emissions, and dust. Prolonged exposure to these sources, all of which produce PM of varying sizes, can affect long-term health, including physiological, psychological, and neurological functioning. For example, consider the following.

- Inflammation: Exposure to air pollution can cause inflammation in the brain, which can cause cognitive impairment [14,15].
- Oxidative stress: Exposure to air pollution can increase oxidative stress, leading to cell damage and cognitive impairment [14,16].
- Reduced oxygen supply: Air pollution can reduce the amount of oxygen available to the body, which can lead to fatigue, decreased endurance, and impaired cognitive function [17–20].
- Increased respiratory effort: Air pollution can increase the effort required to breathe, leading to reduced exercise capacity and decreased performance [18,21,22].
- Neurotransmitter disruption: Exposure to environmental pollutants such as lead, mercury, and polychlorinated biphenyls (PCBs) can alter neurotransmitter function and cause cognitive problems [23,24].
- Epigenetic modifications: Exposure to environmental pollutants can lead to changes in DNA methylation and other epigenetic changes, which can contribute to cognitive problems [25–28].
- The breakdown of the blood–brain barrier: Exposure to air pollution can disrupt the blood–brain barrier, allowing pollutants to enter the brain and cause neurological damage [16].

- Neurotoxicity: Exposure to certain environmental pollutants, such as lead, mercury, and polychlorinated biphenyls (PCBs), can be neurotoxic and affect the nervous system [24,29].

$CO_2$ exposure has been associated with cognitive problems [30–32] and physiological changes in lung and cardiovascular function [33]. Long-term exposure to $NO_2$, which is a gaseous pollutant, has been associated with cardiovascular disease, lung cancer, and respiratory problems, modifying the severity of asthma [34–37]. The inhalation of regulated nitric oxide (NO) under controlled conditions and medications that produce nitric oxide have a wide range of therapeutic uses, such as cardiopulmonary conditions [38,39]. On the other hand, NO, when inhaled in excess amounts, can react with oxygen to form $NO_2$ in the lungs, creating lung problems [39,40]. A higher concentration of NO is considered toxic, although limited studies have been performed on the direct effects of NO inhalation.

In this study, we combined data sets obtained from two different experimental paradigms and provided an overview of our previous work in which biometric data from participants were used to estimate and understand the effects of inhaled ambient $PM_{2.5}$ [41], $CO_2$ [42], and $NO_2$ [43] on the human body using machine learning models and now including $PM_1$ and NO in the study as well. While long-term exposure to air pollution can result in plenty of health-related effects, as mentioned before, short-term exposure to air pollution also has immediate effects on the human body, bringing physiological changes immediately. In this study, we examined the autonomous responses on small temporal ($\sim$2 s) and spatial ($\sim$2 m) scales of the five mentioned pollutants within microenvironments. To comprehensively capture cognitive and physiological changes brought upon by air pollution, we made use of several sensors to measure as many biological measurements as possible, which included skin temperature, respiration rate, blood oxygen saturation ($SpO_2$), heart rate, the galvanic skin response (GSR), the pupil diameter of the left eye, the pupil diameter of the right eye, the distance between the pupils, and the measurement of electrical activity in the brain and heart using electroencephalography (EEG) and electrocardiogram (ECG), respectively.

Since the relations between and among variables are not always linear or functional, we made use of machine learning algorithms to perform regression for nonlinear, non-parametric, multidimensional data. The use of machine learning models has been shown to estimate ambient PM with high degrees of precision, especially $PM_{2.5}$ [44–46]. By simultaneously measuring biological parameters and air quality components, we examined the interaction between the body and the environment while also testing the accuracy of estimating pollutants using machine learning techniques.

## 2. Materials and Methods

The core methodology in the study of these pollutants from two different experimental paradigms was essentially the same; in it, several biometric data (or biometric variables, predictor variables, or cognitive and physiological changes) of participants were collected simultaneously using a biometric suite when a participant was cycling, while other sensors simultaneously measured the ambient air pollutants (or target variable).

### 2.1. Experimental Paradigms

Figure 1 shows the two experimental setup scenarios for the simultaneous measurement of biometric variables and ambient air pollutants. Figure 1a shows the scenario in which a participant wore a biometric suite to capture autonomous changes in the body when cycling a static bike indoors, while sensors placed nearby measured ambient $PM_{2.5}$ and $PM_1$. Similarly, Figure 1b shows the procedure for data collection in which a participant, equipped with the same biometric suite to measure biometric variables, was cycling outdoors with an electric car behind them that was equipped with several different sensors measuring ambient $CO_2$, $NO_2$, NO, $PM_1$, and $PM_{2.5}$ simultaneously. An electric car was used so that the sensors placed in the trunk of the car did not measure the pollutants produced by the car but only those of the ambient air.

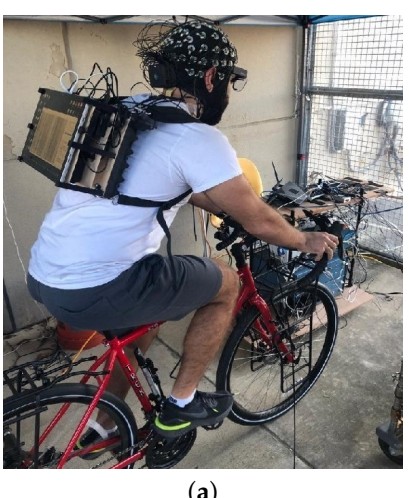 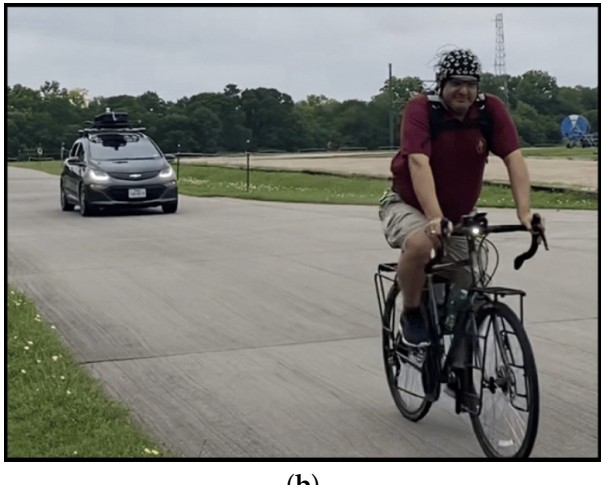

(**a**)                      (**b**)

**Figure 1.** Two of the experimental paradigms for biometric and environmental data collection in which a participant is wearing the same biometric suite for biometric data collection. (**a**) Each of the participants rode a static bike with sensors placed nearby to measure ambient $PM_{2.5}$ and $PM_1$. (**b**) The participants in the study rode a bicycle followed by an electric car measuring environmental $CO_2$, $NO_2$, and $NO$ among other environmental variables. Source: Figure 4 from [46].

Two of the experimental paradigms in this study shared some similarities and differences as well. Table 1 shows some of the similarities between the two experimental paradigms.

**Table 1.** Similarities between the two experimental paradigms.

| Similarities |
| :---: |
| Use of the same biometric suite to measure biometric variables. |
| Pollutants were measured using sensors that were in close proximity to the participant. |
| Machine learning models were used to estimate the inhaled pollutants and examine the autonomous responses in the human body. |

The differences between the two experimental paradigms are shown in Table 2.

**Table 2.** Differences between the two experimental paradigms.

| Bike in Motion | Static Bike Ride |
| :---: | :---: |
| A single participant for data collection. | Multiple participants for data collection. |
| The participants rode a bike on multiple tracks. | The participants rode a stationary bike. |
| The location of the data collection was outdoors in Breckenridge Park in Richardson. | The location of the data collection was indoors inside the WSTC building at the University of Texas at Dallas, Richardson. |
| In this study, the measurement of ambient $CO_2$, $NO_2$, and $NO$ as an environmental variable was considered. | The measurement of $PM_1$ and $PM_{2.5}$ as an environmental variable was considered. |
| Data collection was carried out in 2021. | Data collection took place in 2021 and 2022. |
| All of the 64 electrodes on the EEG headset were working. | The T7 electrode of the EEG headset was not working. |

### 2.2. Data Collection

The process of data collection for both of the experimental paradigms involved the simultaneous measurement of biometric data using the same biometric suite and environ-

mental data (or the target variable or pollutant). Several biometric variables were measured, among which the ones that were considered for the study are presented in Table 3.

The EEG data were collected using a Cognionics EEG headset consisting of 64 electrodes following the 10–10 nomenclature system [47] (https://www.cgxsystems.com/mobile-128, accessed on 16 January 2024) with a sampling rate of 500 Hz. Among the rest of the physiological responses (or non-EEG variables) the ECG, GSR, $SpO_2$, respiration rate, skin temperature, and heart rate were measured using a Cognionics AIM Generation 2 device (https://www.cgxsystems.com/auxiliary-input-module-gen2, accessed on 16 January 2024) with a sampling rate of 500 Hz. The Tobii Pro Glasses 2 (https://www.tobii.com/products/discontinued/tobii-pro-glasses-2, accessed 16 January 2024) provided several pupillometric measurements at a sampling rate of 100 Hz, but the ones that were considered were the pupil diameter of the left eye, the pupil diameter of the right eye, and the distance between the pupils.

**Table 3.** List of biometrics measured in both experiments.

| Biometric Variable | Units | Location of the Sensor |
| --- | --- | --- |
| Electroencephalography (EEG) | Volt (V) | A headset |
| Electrocardiography (ECG) | Volt (V) | Upper part of chest |
| Galvanic Skin Response (GSR) | MicroSiemens (μSiemens) | Upper back |
| Oxygen Saturation ($SpO_2$) | Percentage (%) | Left ear |
| Respiration rate | Breathing rate per minute (brpm) | Same device used to measure GSR |
| Skin temperature | °C | Right temple |
| Heart rate | Beats per minute (bpm) | Same device used to measure $SpO_2$ |
| Pupil diameter of both eyes | Millimeter (mm) | Eye tracking glasses |
| Distance between pupils | Millimeter (mm) | The same eye tracking glasses |

The data obtained from each of the 64 electrodes (or channels) of the EEG headset were received as a time series of voltages. These voltages were measured with respect to a virtual reference that was averaged from all the channels. The voltage time series could be transformed from the time domain to the frequency domain. One of the ways to do so is the Welch method [48], which was implemented using scipy [49]. The transformation thus identified a power spectrum density ($V^2/Hz$) in the Y-axis and frequency in the X-axis. The frequency could be divided into five frequency bands named delta, theta, alpha, beta, and gamma, each representing a different brain state. With the data obtained from each of the 64 electrodes, transforming each into a frequency domain and dividing each frequency into five frequency bands provided a total of 320 biometric variables from the EEG headset.

From the three measured pupillometric variables, other variables such as the average pupil diameter of the two pupils, the difference between pupil diameters of the left and right eyes, and the absolute value of the difference between the pupil diameters were calculated, providing extra features that were considered.

Before data collection for the study began, in each of the experiments, baseline biometric measurements were made for two minutes with the participants' eyes closed and their eyes open. The biometric suite was placed on the participants so that it had little effect on their physiological responses.

$CO_2$ measurement was performed using the LI-COR LI-850 device (https://www.licor.com/env/support/LI-850/topics/description.html#Onlineresources, accessed 21 January 2024) with a sampling rate of 0.5 Hz (twice every second). The measurement of $NO_2$ and NO was carried out using the Model 405 nm $NO_2$/NO/$NO_x$ Monitor from 2B technologies (https://2btech.io/items/other-monitors/model-405-nm-no2-no-nox-monitor/, accessed 21 January 2024) with a sampling rate of 0.2 Hz (once every 5 s), and the measurement of $PM_1$ and $PM_{2.5}$ was carried out using the Fidas Frog device (https://www.palas.de/

en/product/fidasfrog, accessed 21 January 2024) with a sampling of 1 Hz (once every 1 s). The measurement of biometric data was stopped when cycling was stopped and collected again when cycling was resumed.

At times, the precision of the data captured via biometric sensors can be compromised due to their movement, resulting in the possibility of no values being recorded. Furthermore, the devices also have different sampling rates. Therefore, the data were cleaned and down-sampled to 1 s for $CO_2$, 5 s for $NO_2$, 5 s for NO, and 1 s for $PM_{2.5}$. The total number of biometric variables used and the number of data records collected for each pollutant are shown in Table 4.

**Table 4.** Collection of data for five pollutants.

| Pollutant | Total Number of Biometrics | Days of Data Collection | Number of Trials | Data Records in Each Trial | Total Number of Data Records |
|---|---|---|---|---|---|
| $CO_2$ | 329 | 2 | 4 | 710, 696, 673, 238 | 2317 |
| $PM_{2.5}$ | 322 | 4 | 4 | 298, 239, 528, 318 | 1383 |
| $PM_1$ | 322 | 4 | 4 | 298, 239, 528, 318 | 1383 |
| $NO_2$ | 329 | 3 | 6 | 136, 23, 126, 120, 132, 45 | 582 |
| NO | 329 | 3 | 6 | 81, 15, 96, 88, 98, 32 | 410 |

The data collection for $CO_2$, $NO_2$, and NO was carried out on three separate days: Of 26 May, 9 June, and 10 June 2021, accurate data for $CO_2$ readings were collected only on 9 June 2021 and 10 June 2021 with 2 trials on each day. Accurate data for $NO_2$ and NO were received on all 3 days with 2 trials on each day. Data collection for $PM_1$ and $PM_{2.5}$ took place on October 21 2021, January 14 2022, January 27 2022, and February 9 2022 with different participants on each day.

The data obtained for $NO_2$ and NO from the measuring device were filtered to include only records that passed multiple quality criteria. These quality criteria included (a) the cell flow rate of the sample gas between (1400 to 1600) cc/min, (b) the ozone flow rate between (60 to 80) cc/min, (c) the cell photodiode voltage (PDV) of at least 0.6 Volts, and (d) the PDV ozone generator of at least 0.1 Volts.

*2.3. Data Analysis and Developing a Machine Learning Model*

After the construction of the four datasets, each consisting of biometric variables as the input features and one output target variable, we sought to estimate the inhaled pollutant concentrations. Each target variable was estimated separately using random forests [50] for multidimensional, nonlinear regression using the ensemble Random Forest Regressor package from scikit-learn (version 1.0.2) [51] in Python (version 3.11.1). All models were trained using 80% of the data, and the remaining 20% were used as an independent test set. The determination coefficient ($R^2$) and the root mean square error (RMSE) were calculated between the true values of the pollutant and the estimated values of the pollutant to quantify the goodness of fit. Scatter plots, quantile–quantile plots, and time series plots of the actual and estimated pollutant values were also plotted for a qualitative analysis of the goodness of fit.

Each of the scatter diagrams was overlaid with a 1:1 black line to indicate how far the prediction was from the true values with data points with an exact prediction lying on the 1:1 line. A quantile–quantile plot for each of the four machine learning models was drawn and overlaid with percentiles to indicate where in the distribution the data points deviated from the actual values with data points that had an exact prediction lying on the red 1:1 line.

To identify the effectiveness of biometric variables in predicting the target variable, the SHAP values (SHapley Additive exPlanations) [52,53] of the SHAP library (version 0.41.0)

were used to rank the predictor variables in descending order. The SHAP values for variables below the ninth order were found to be small and, thus, less effective in making the prediction. Since the data were mostly nonlinear, the top 9 of those variables in the predictor ranking were then used, and a 10 × 10 mutual information matrix including the pollutant to be estimated was calculated using a package from scikit-learn [51] to identify the nonlinear relationship between the variables. These mutual information values were greater than zero, with higher values indicating a stronger relationship and zero values indicating that the two variables were independent of each other.

## 3. Results

Among the several biometric variables that were measured in this study, some of the readings were not easily measurable, and the devices were expensive as well, for example, the EEG and Tobii Pro glasses 2. Other variables, such as skin temperature, $SpO_2$, heart rate, respiration rate, GSR, and ECG, could be measured relatively easily and were also inexpensive. Therefore, the study was classified into two parts; first, we considered all the biometric features that were either measured or calculated, and second, we considered biometric variables that could be easily measured and were accessible.

### 3.1. Using all Features

Table 5 shows the coefficient of determination ($R^2$) and RMSE between the true values of the pollutant and the estimated values of the pollutant in the training set and the testing set for each pollutant using the Ensemble Random Forest Regressor package from scikit-learn and considering all biometric variables with default hyperparameters.

**Table 5.** Quantification of the estimation of the pollutant using all features with default hyperparameters of the random forest algorithm.

| Pollutant | Train $R^2$ | Test $R^2$ | Train RMSE | Test RMSE | Number of Biometrics Inputs |
|-----------|-------------|------------|------------|-----------|------------------------------|
| $PM_1$ | 0.99 | 0.99 | 0.03 µg/m$^3$ | 0.07 µg/m$^3$ | 322 |
| $CO_2$ | 0.99 | 0.97 | 9.89 ppm | 21.63 ppm | 329 |
| $PM_{2.5}$ | 0.99 | 0.97 | 0.14 µg/m$^3$ | 0.37 µg/m$^3$ | 322 |
| NO | 0.96 | 0.45 | 5.27 ppb | 11.50 ppb | 329 |
| $NO_2$ | 0.91 | 0.12 | 2.95 ppb | 8.93 ppb | 329 |

Table 5 shows that the train $R^2$ for all pollutants was nearly 1, and the train RMSE was also low, which was expected since this part of the data set was used in the machine learning model for learning. The independent test $R^2$ for $PM_1$, $CO_2$, and $PM_{2.5}$ was also almost 1, and the RMSE was also small, indicating that the performance and generalization of the machine learning model in estimating $PM_1$, $CO_2$, and $PM_{2.5}$ was very good. For pollutants such as $NO_2$ and NO, for which we had far fewer data records, the performance was not as good, with low test $R^2$ values. One possible explanation for the result is that there were not enough training examples, explained detailly in Sections 3.1.2 and 3.1.3 for $NO_2$ and NO respectively. The $R^2$ values and the RMSE values for all these pollutants can change according to the way the data are shuffled. For $PM_1$, $PM_{2.5}$, and $CO_2$, these values remained close to each other because there was an abundance of data points over a range of values. However, for $NO_2$ and NO, these values changed to some extent, depending on how the data were shuffled, especially considering the large number of predictor variables for a relatively small data set. When the algorithm was run five times, the average $R^2$ between true and estimated values of $NO_2$ in the training set and the test set was 0.91 and 0.12, respectively, and the average RMSE between the true and estimated values of $NO_2$ was 3.11 ppb and 8.56 ppb in the training set and test set, respectively. Similarly, for NO, when the algorithm was run five times, the average $R^2$ value in the training and testing set was 0.93 and 0.21, respectively, and the average RMSE was 5.23 ppb and 11.62 ppb, respectively.

To test whether the precision of the estimation could be improved further, we optimized the hyperparameters of the random forest algorithm. Two of the hyperparameters were optimized, including (a) *n_estimators* (the total number of trees), and (b) *max_features* (the number of features considered at each split). To find the best combination of these two parameters, scikit-Learn's *GridSearchCV* was used. A set of integers of these two parameters was provided as an input to the *GridSearchCV* function, which then ran all the possible combinations with K-fold cross-validation. With the provided set of *n_estimators* (say m), a set of *max_estimators* (say n), and K- fold cross-validation, the model was trained a total of m × n × k times, and *GridSearchCV* returned the best possible combination that had the lowest mean squared error. Since the default value of the *n_estimators* parameter is 100, we provided *n_estimators* with a set of values that were either less than 100 or greater than or equal to 100. Also, the default value of *max_features*=1.0, and it considered features equal to the total number of features when considering the best split. Thus, a set of integers less than the total number of features was provided to the *max_features* parameter. The number of features considered at each split was then equal to the number in the set of values provided to the *max_features* parameter.

Table 6 shows the best combination of the *n_estimators* and the *max_features* parameter for each of the pollutants. Since the best value of the parameter was between the set of values provided to the *GridSearchCV* function, other values did not need to be tested.

**Table 6.** Set of hyperparameters provided as inputs to *GridSearchCV* and the best possible parameter when considering all available features.

| Pollutant | Set of *n_estimators* | Set of *max_features* | Folds for Cross-Validation | Total Number of Training | Optimized Parameter |
|---|---|---|---|---|---|
| $PM_1$ | 80, 90, 100, 110, 120 | 250, 275, 300 | 3 | 45 | 110, 275 |
| $CO_2$ | 80, 90, 100, 110, 120 | 250, 275, 300, 325 | 3 | 60 | 110, 300 |
| $PM_{2.5}$ | 100, 110, 120, 140 | 150, 180, 200, 250, 275 | 3 | 60 | 120, 200 |
| NO | 80, 90, 100, 110, 120 | 250, 275, 300, 325 | 3 | 60 | 110, 275 |
| $NO_2$ | 80, 90, 100, 110, 120 | 180, 200, 240, 250 | 3 | 60 | 90, 200 |

After the hyperparameters of the random forest algorithm were optimized, the model was then used to test the accuracy of the prediction. Table 7 shows the quantification of the estimation of the pollutants with optimized hyperparameters.

**Table 7.** Quantification of the estimation of the pollutants using all features with optimized hyperparameters of the random forest algorithm.

| Pollutant | Train $R^2$ | Test $R^2$ | Train RMSE | Test RMSE | Number of Biometrics Inputs |
|---|---|---|---|---|---|
| $PM_1$ | 0.99 | 0.99 | 0.03 μg/m$^3$ | 0.07 μg/m$^3$ | 322 |
| $CO_2$ | 0.99 | 0.98 | 10.37 ppm | 21.61 ppm | 329 |
| $PM_{2.5}$ | 0.99 | 0.97 | 0.15 μg/m$^3$ | 0.37 μg/m$^3$ | 322 |
| NO | 0.96 | 0.37 | 5.33 ppb | 9.29 ppb | 329 |
| $NO_2$ | 0.91 | 0.11 | 2.95 ppb | 10.08 ppb | 329 |

When comparing Tables 6 and 7, we observe that the results were almost similar when the hyperparameters were optimized. For NO, the results seem to have decreased. Since Pearson's correlation coefficient is highly susceptible to outliers, and the data are shuffled each time the algorithm is run, this disparity was expected. We ran the algorithm five times for $NO_2$ and NO. The average RMSE between the true and estimated values of $NO_2$ was found to be 3.26 ppb and 6.45 ppb in the training and testing set, respectively, while the $R^2$ was found to be 0.92 and 0.19. Similarly, the average RMSE between the true and estimated values of NO was found to be 5.01 ppb and 12.83 ppb in the training and testing set, respectively, while the $R^2$ was found to be 0.93 and 0.15.

### 3.1.1. Carbon Dioxide

For the study of $CO_2$, a total of 329 biometric input variables were taken into account, including the 320 variables from the EEG data, and the remaining variables included the following: ECG, respiration rate, $SpO_2$, heart rate, GSR, skin temperature, pupil distance, the average pupil diameter, and the absolute value of the difference between pupil diameters. Figure 2a shows a SHAP value beeswarm plot of the top nine features in descending order to indicate the biometric variables that were the most influential in estimating $CO_2$. Figure 2b shows a mutual information matrix consisting of the nine variables with the nine highest SHAP values and $CO_2$.

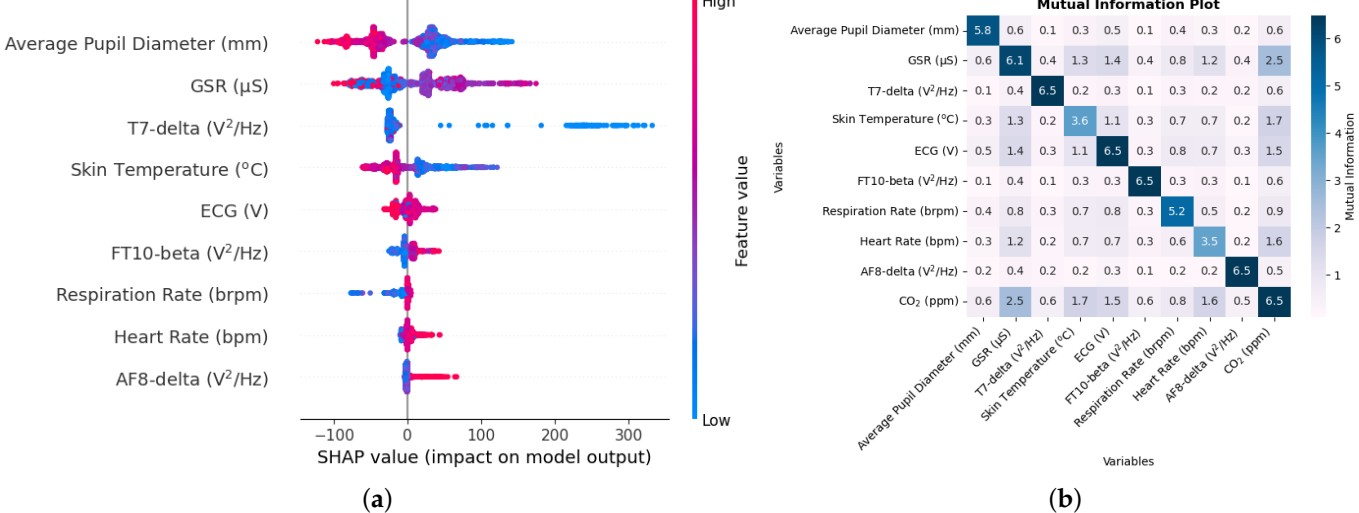

(a)  (b)

**Figure 2.** (**a**) A SHAP value beeswarm plot of the top nine features in descending order for estimating the inhaled $CO_2$. (**b**) A mutual information matrix consisting of the top nine biometric variables that were the most influential in the prediction of $CO_2$ and the target variable, $CO_2$.

These SHAP values for $CO_2$ on the X-axis are expressed in ppm units. As indicated by the SHAP values, the average diameter of the pupil, the GSR, and skin temperature were among the top physiological responses that were the most effective in predicting $CO_2$. The order of these variables can change, depending on how the data are shuffled, especially when the SHAP values are close to each other—for example, for features in order numbers 5, 6, and 7. The plot also indicates that higher values of the average pupil diameter tend to decrease the prediction, while lower values tend to increase the prediction, as large portions of SHAP values for the average pupil diameter are negative and positive, respectively.

In addition to the fact that the diameter of the pupil changes, depending on the light entering it, the diameter of the pupil has been associated with cognitive ability [54]; as mentioned before, $CO_2$ intake is linked to cognitive issues [30–32] as well. The GSR sensor measures the response to sweat, and sweating can be caused by physical tasks such as cycling. $CO_2$ inhalation can cause sweating when the concentration is 6 to 10% [55]. Other biometric variables include the respiration rate, heart rate, and skin temperature, and considering that exposure to $CO_2$ can cause physiological changes in lung and cardiovascular function [33], it was expected that these variables would be affected by $CO_2$ intake.

Similarly, the EEG variables included T7, FT10, and AF8 electrodes with frequency band delta, beta, and delta bands, respectively. According to the 10–10 system of nomenclature [47], electrodes with odd numbers are on the left side, and those with even numbers are on the right side. The T7 electrode is above the temporal lobe, which is associated with speech and short-term memory [56]. The FT10 electrode is located between the frontal and temporal lobes. The SHAP value of the AF8 electrode was very low, and therefore, all variables below the order had a smaller SHAP value and provided a small contribution to $CO_2$ prediction.

The mutual information matrix shows that inhaling $CO_2$ has a high nonlinear relationship with the GSR, skin temperature, ECG, respiration rate, and heart rate, indicating the several changes brought about via $CO_2$ intake. Similarly, these biometrics are also mutually related to each other, as GSR has high mutual information with the average pupil diameter, skin temperature, ECG, and heart rate, while skin temperature has high mutual information with ECG, the ECG with the respiration rate, and the respiration rate with the heart rate.

The scatter diagram and the quantile–quantile plot for $CO_2$ are shown in Figure 3. The scatter diagram in Figure 3a shows that not only most of the data points in the training set but also, more importantly, those in the testing set lie very close to the black 1:1 line, indicating that the predictions are close to each other for a large portion of the dataset.

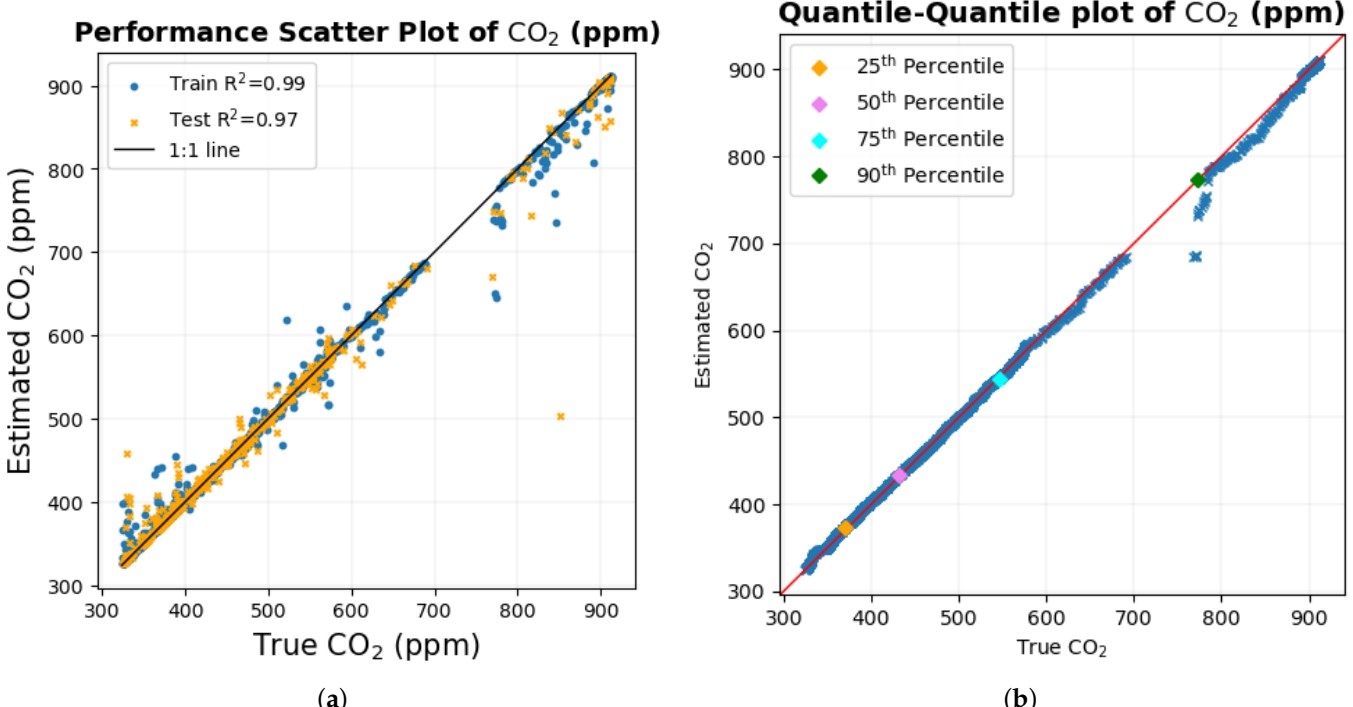

**Figure 3.** (**a**) Scatter diagram of the true values of $CO_2$ against the estimated values of $CO_2$ with a black 1:1 line overlaid. (**b**) Quantile–quantile plot of the true values of $CO_2$ against the estimated values of $CO_2$ with a red 1:1 line overlaid.

The quantile–quantile plot in Figure 3b also shows that, for most of the distribution, the data points lie close to the red 1:1 line. The quantiles in the distribution deviate for values between 700 and 800 ppm, and one of the possible reasons could be the scarcity of data points in this range of value, which is also depicted in the scatter diagram.

3.1.2. Nitrogen Dioxide

The 329 variables that were considered for the study of $NO_2$ include the 320 EEG variables, ECG, respiration rate, $SpO_2$, heart rate, GSR, skin temperature, average pupil diameter, pupil distance, and difference in pupil diameter.

In the case of $NO_2$, the estimate was not as good, as indicated by the value of $R^2$ and RMSE between the true and estimated values of $NO_2$ in Table 5. However, Figure 4a shows the SHAP value beeswarm plot of the top nine biometric features that were the most influential in estimating $NO_2$. Figure 4b shows the mutual information matrix of the top nine features chosen by SHAP values and $NO_2$.

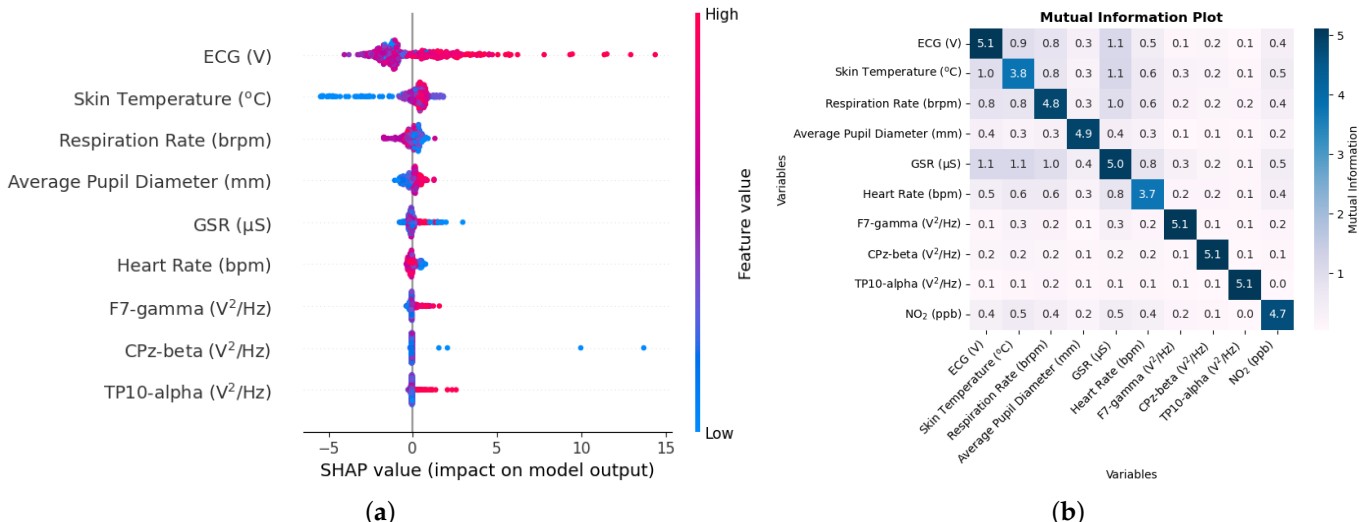

**Figure 4.** (**a**) A SHAP value beeswarm plot of the top nine features in descending order that were useful in estimating $NO_2$. (**b**) Mutual information matrix consisting of the top nine biometric variables that were the most influential in predicting $NO_2$ and the target variable, $NO_2$.

The SHAP values, in this case on the X-axis, are in units of ppb. The SHAP value of the ECG and skin temperature is relatively higher than the other variables, so these variables do not tend to change order. However, the ordering of the rest of the variables can change, depending on how the data are shuffled, as the SHAP values are close to each other, especially at the lower end of the order. The plot also shows that lower values of skin temperature tend to decrease the prediction, while higher values tend to increase the prediction. As long and short-term exposure to $NO_2$ is associated with cardiovascular disease [57], it is likely that the ECG is one of the main variables. Since the inhalation of a higher concentration of $NO_2$ causes inflammation of the airways, changes in the respiration rate, skin temperature, and sweating are also likely to affect the GSR sensor.

Other variables include EEG ones. The F7 electrode is one of the main EEG variables. The SHAP value of the F7-gamma variable and the following two variables are low compared to the rest of the variables, indicating their low effectiveness in estimating $NO_2$.

The mutual information matrix in Figure 4b shows that $NO_2$ has high mutual information with the ECG, skin temperature, heart rate, and GSR, which was again to be expected, as the SHAP values for these variables were high. The matrix also shows that the ECG has higher mutual information with skin temperature, respiration rate, heart rate, and GSR, while skin temperature has higher mutual information with the respiration rate, heart rate, and GSR, the respiration rate has higher mutual information with the heart rate and GSR, and the heart rate has higher mutual information with the GSR. This is similar to what is seen in the mutual information matrix in Figure 2b, indicating that the variables are mutually related to each other.

The scatter plot and the quantile–quantile plot for $NO_2$ are shown in Figures 5a and 5b, respectively. The scatter diagram in Figure 5a shows that the lower values of the data points lie close to the black 1:1 line, where there is an abundance of data. The quantile–quantile graph in Figure 5b shows that around 90% of the data are less than 20 ppb, where the quantile–quantile graph is close to the red 1:1 line. As the values of $NO_2$ increase, the number of data points is low; this could possibly have caused the number of data points to deviate from the 1:1 black and red line for higher $NO_2$ values, as there is a very small number of data points from which the machine learning model can learn.

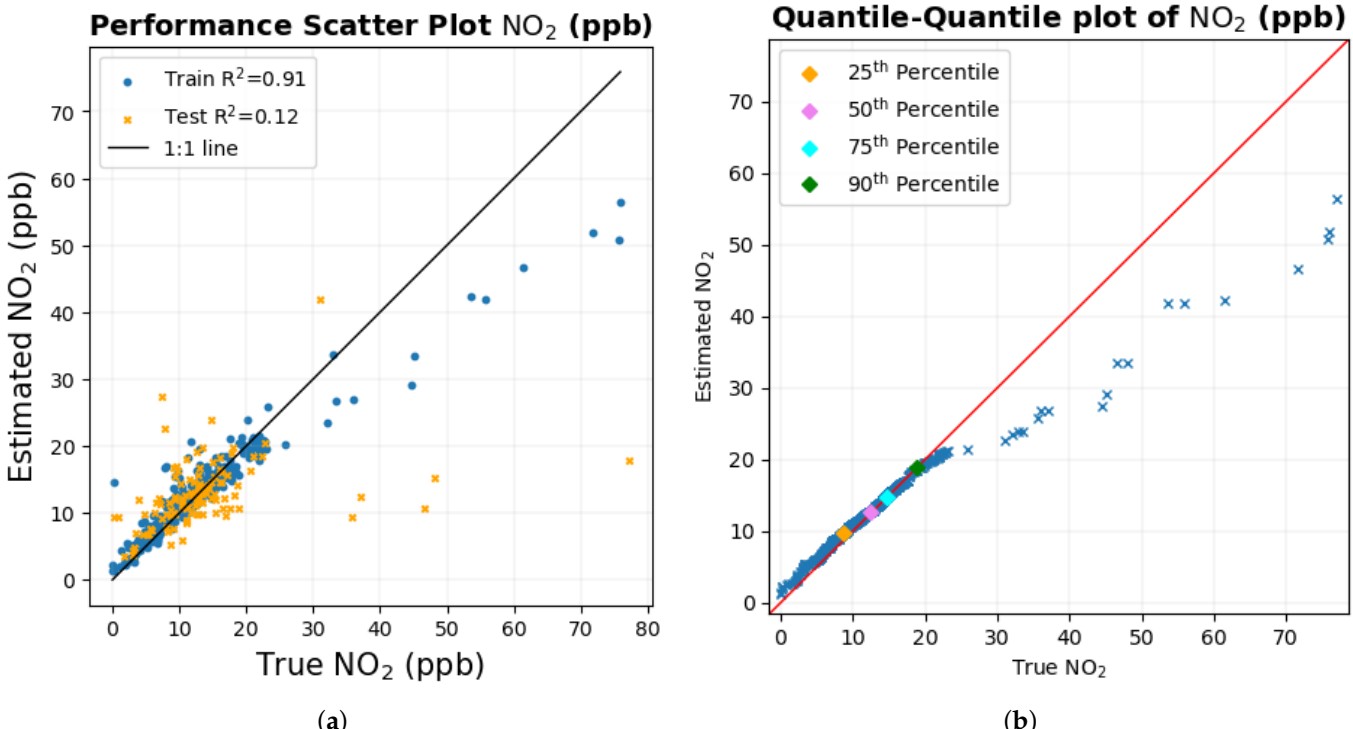

**Figure 5.** (**a**) Scatter diagram of the true values of $NO_2$ against the estimated values of $NO_2$ with a black 1:1 line overlaid. (**b**) Quantile–quantile plot of the true values of $NO_2$ against the estimated values of $NO_2$ with a red 1:1 line overlaid.

### 3.1.3. Nitric Oxide

In the NO study, the same 329 variables used in $NO_2$ were considered. Similar to the case of $NO_2$, the estimation of NO using biometrics did not appear to be very accurate, as indicated by $R^2$ and the RMSE values between the true and estimated values of NO in Table 5.

The SHAP value beeswarm plot in Figure 6a shows that physiological responses such as skin temperature, average pupil diameter, and ECG are among the main biometric variables in estimating NO. The unit of the SHAP value on the X-axis here is ppb. The plot also shows that higher skin temperature values tend to lower the prediction, while lower values tend to increase the prediction. Since the inhalation of NO when reacted with oxygen can create $NO_2$, skin temperature and ECG were possibly affected, and they were also common variables in $NO_2$. There appears to be a large number of EEG variables as well. The PO7 electrode is located between the parietal and occipital lobes on the left side of the brain. The gamma band that seems common is dominant in tasks that involve concentration [58]. Other biometric variables, such as Fp2-gamma, Fpz-beta, and those below them, have low SHAP values and, thus, provide less of a contribution to NO estimation.

The mutual information matrix in Figure 6b shows that, among the predictor variables, NO has high mutual information with skin temperature and ECG. Skin temperature and ECG also have high mutual information with each other.

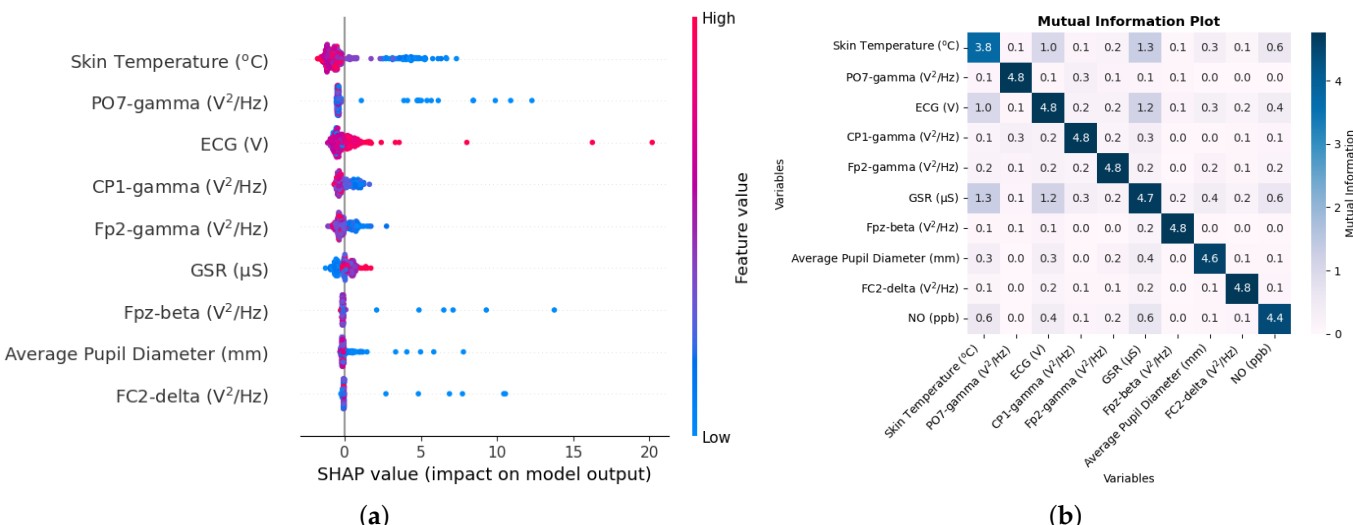

(**a**)                                                                                    (**b**)

**Figure 6.** (**a**) A SHAP value beeswarm plot of the top nine features in descending order that were useful in estimating NO. (**b**) Mutual information matrix consisting of the top nine biometric variables that were the most influential in predicting NO and the target variable, NO.

Figure 7 shows the scatter plot and the quantile–quantile plot of the true values of NO compared to the estimated values of NO.

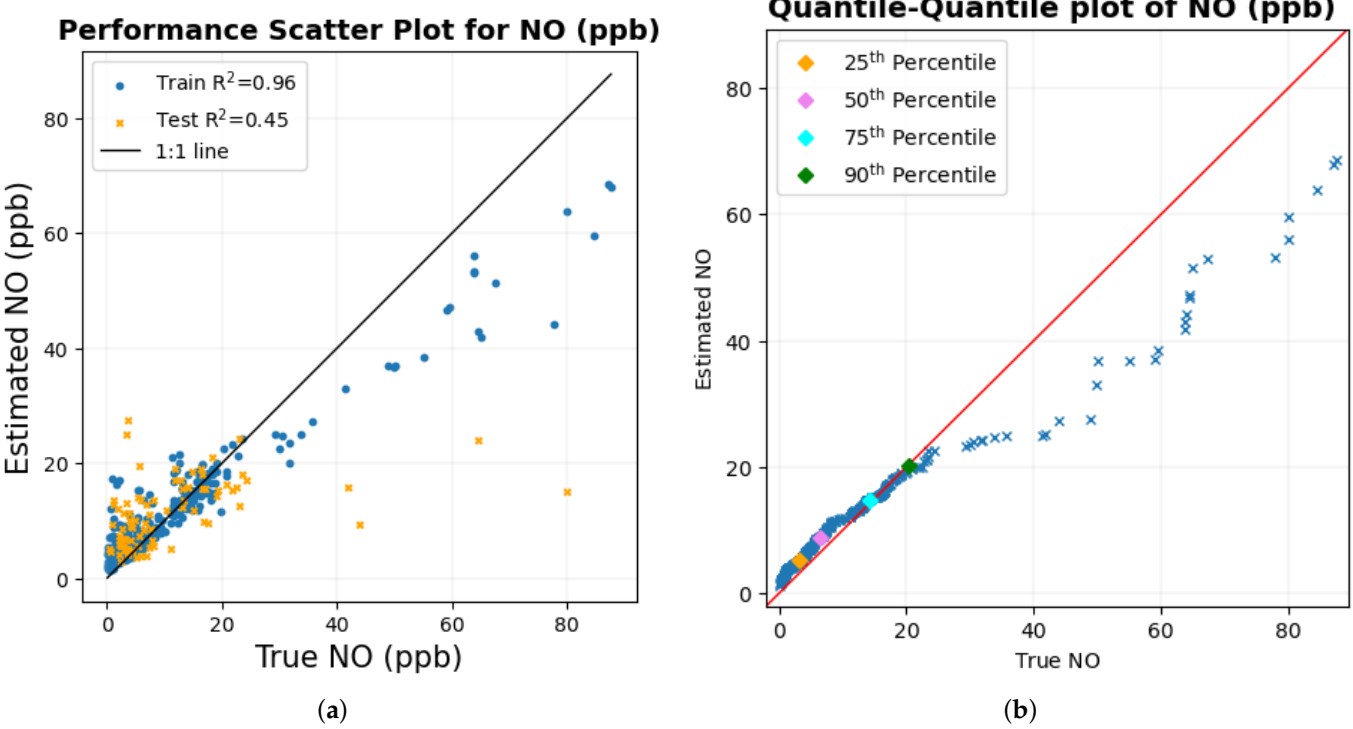

(**a**)                                                                                    (**b**)

**Figure 7.** (**a**) Scatter diagram of the true values of NO against the estimated values of NO with a black 1:1 line overlaid. (**b**) Quantile–quantile graph of the true values of NO against the estimated values of NO with a red 1:1 line overlaid.

The overall structure of the scatter diagram and the quantile–quantile graph for NO looks similar to that of $NO_2$, with smaller values of NO lying close to the corresponding 1:1 line, where there is an abundance of data points. The quantile–quantile plot in Figure 7b shows that more than 90% of the data are below around 20 ppb. As the values of NO get

larger, these data points tend to deviate from the 1:1 line; one of the possible reasons could be the scarcity of data points in the region.

### 3.1.4. $PM_1$

Since, in the static bike ride experimental setup in which the measurement of $PM_1$ and $PM_{2.5}$ was performed, the T7 electrode of the EEG headset did not give any readings, the number of biometric variables was reduced to 322. These biometric variables include the 315 variables of the EEG headset, respiration rate, $SpO_2$, heart rate, skin temperature, the average pupil diameter, pupil distance, and the difference in pupil diameter. As shown in Table 5, the $PM_1$ performance was the highest with an $R^2$ value of 0.99 and the lowest RMSE of 0.07 $\mu g/m^3$ in the test set.

The SHAP value beeswarm plot in Figure 8a shows that skin temperature, pupil distance, and heart rate were among the main features that were the most influential in estimating $PM_1$. Skin temperature and the distance of the pupil were also important variables in the estimation of $PM_1$ when a single participant was used for the study [46]. The distance of the pupils, which indicates the vergence of the eyes, has been associated with the attention load [59]. A series of EEG variables were amongst the top variables in which the SHAP value of the Cz-delta variable was low, and all the variables below it were even lower, very close to zero. Thus, removing these features from the study would have little effect in estimating $PM_1$.

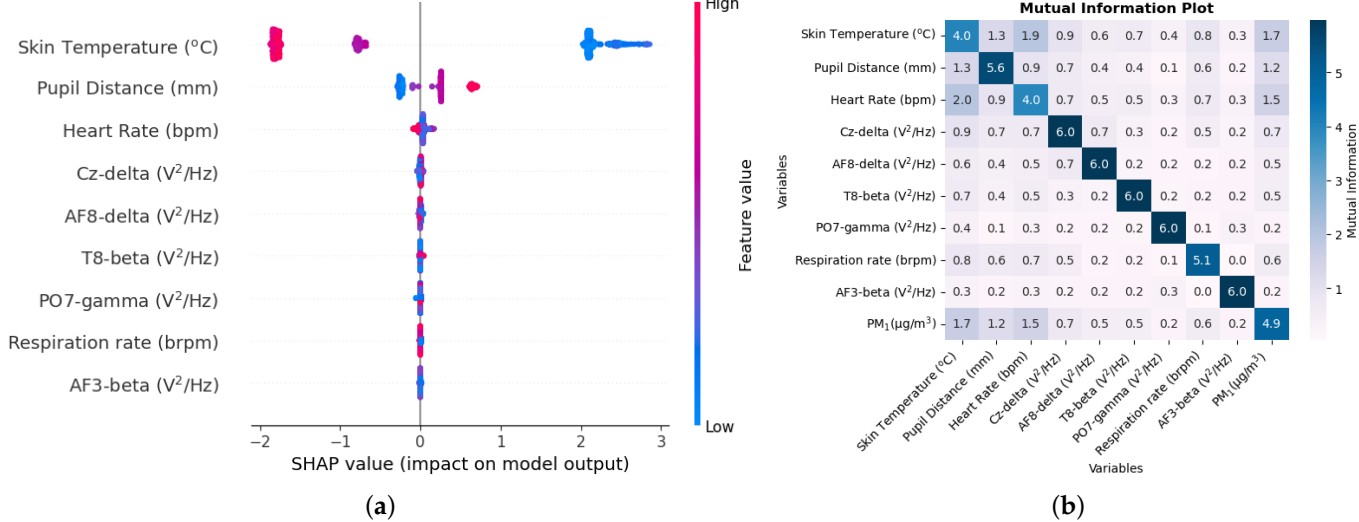

**Figure 8.** (**a**) A SHAP value beeswarm plot of the top nine features in descending order that are useful in estimating $PM_1$. (**b**) Mutual information matrix consisting of the top nine biometric variables that were the most influential in predicting $PM_1$ and the target variable, $PM_1$.

The mutual information matrix in Figure 8b shows that $PM_1$ has high mutual information with physiological responses such as pupil distance, heart rate, and respiration rate, and it also shows that the physiological responses are indeed mutually related to each other.

The scatter diagram and quantile–quantile graph with true $PM_1$ values on the X-axis and estimated $PM_1$ values on the Y-axis are shown in Figure 9a and b, respectively.

Both the scatter plot and the quantile–quantile plot show that the data points are very close to the 1:1 line of the corresponding graph, indicating that the prediction is the most accurate and precise among all the pollutants.

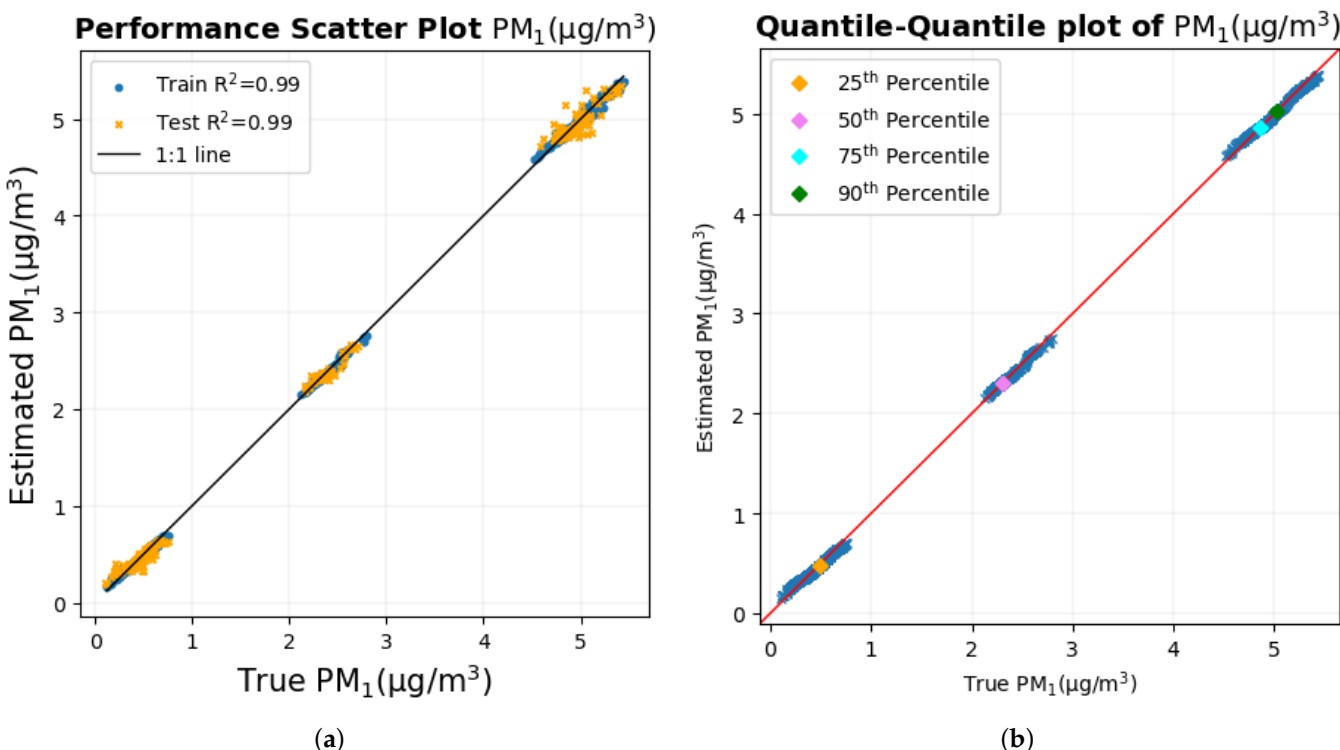

**Figure 9.** (**a**) Scatter diagram of the true values of $PM_1$ against the estimated values of $PM_1$ with a black 1:1 line overlaid. (**b**) Quantile–quantile plot of the true values of $PM_1$ against the estimated values of $PM_1$ with a red 1:1 line overlaid.

### 3.1.5. $PM_{2.5}$

The biometric variables that were considered in the study of $PM_{2.5}$ were the same as those of $PM_1$. As shown in Table 5, the estimation of $PM_{2.5}$ was highly accurate, as indicated by an $R^2$ value between the true and estimated values of $PM_{2.5}$ in both the training and the test set, which was almost 1. The RMSE is also one of the lowest among all the pollutants.

The SHAP value beeswarm plot in Figure 10a shows that the physiological responses that were the most effective in estimating $PM_{2.5}$ included skin temperature, pupil distance, average pupil diameter, and heart rate, three of which are common to that of $PM_1$. The SHAP values in the X-axis here are in $\mu g/m^3$. The inflammatory response created via the higher concentration of $PM_{2.5}$ can possibly cause changes in skin temperature and heart rate. Furthermore, $PM_{2.5}$ also causes adverse health effects on the respiratory system [3], and heart problems [5] could be the reason why the heart rate is one of the most important variables. The size of the pupils has been associated with cognitive ability [54].

Several EEG variables are on the list of the top nine variables. The FT8 electrode is located on the right side of the brain between the frontal and temporal lobes. The CP4-gamma variable has a low SHAP value with variables below the order of even lower SHAP values, indicating that the elimination of these variables will have little effect on the prediction of $PM_{2.5}$.

The mutual information matrix in Figure 10b shows that there is some sort of non-linear relationship between $PM_{2.5}$ and skin temperature, pupil distance, and heart rate, which was again to be expected, as the SHAP values for these variables were also high. Just as in the case of other pollutants for which physiological changes were related to each other, such was the case here with skin temperature, pupil distance, and heart rate mutually being related to each other.

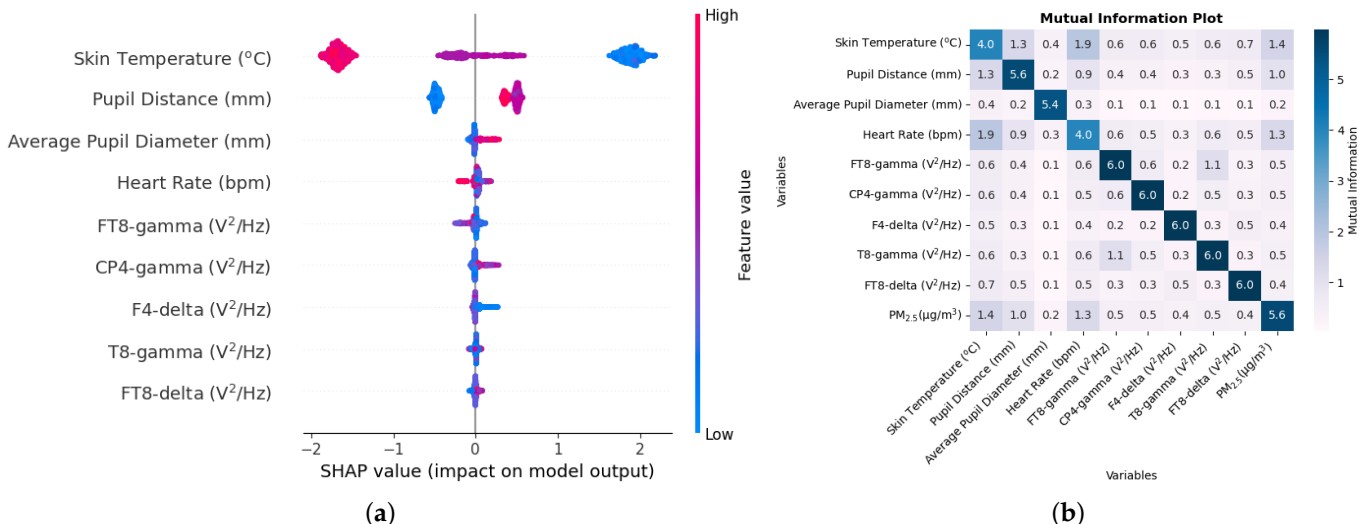

(**a**) (**b**)

**Figure 10.** (**a**) A SHAP value beeswarm plot of the top nine features in descending order that were useful in estimating PM$_{2.5}$. (**b**) Mutual information matrix consisting of the top nine biometric variables that were the most influential in predicting PM$_{2.5}$ and the target variable, PM$_{2.5}$.

Figure 11a,b show the scatter plot and the quantile–quantile plot of the true values of PM$_{2.5}$ versus the estimated values of PM$_{2.5}$. The scatter plot and the quantile–quantile plot in Figure 11a and b, respectively, show that most of the data points lie in the 1:1 line of the corresponding graph. This shows that, for most of the data set, the estimate was close to the true PM$_{2.5}$ values.

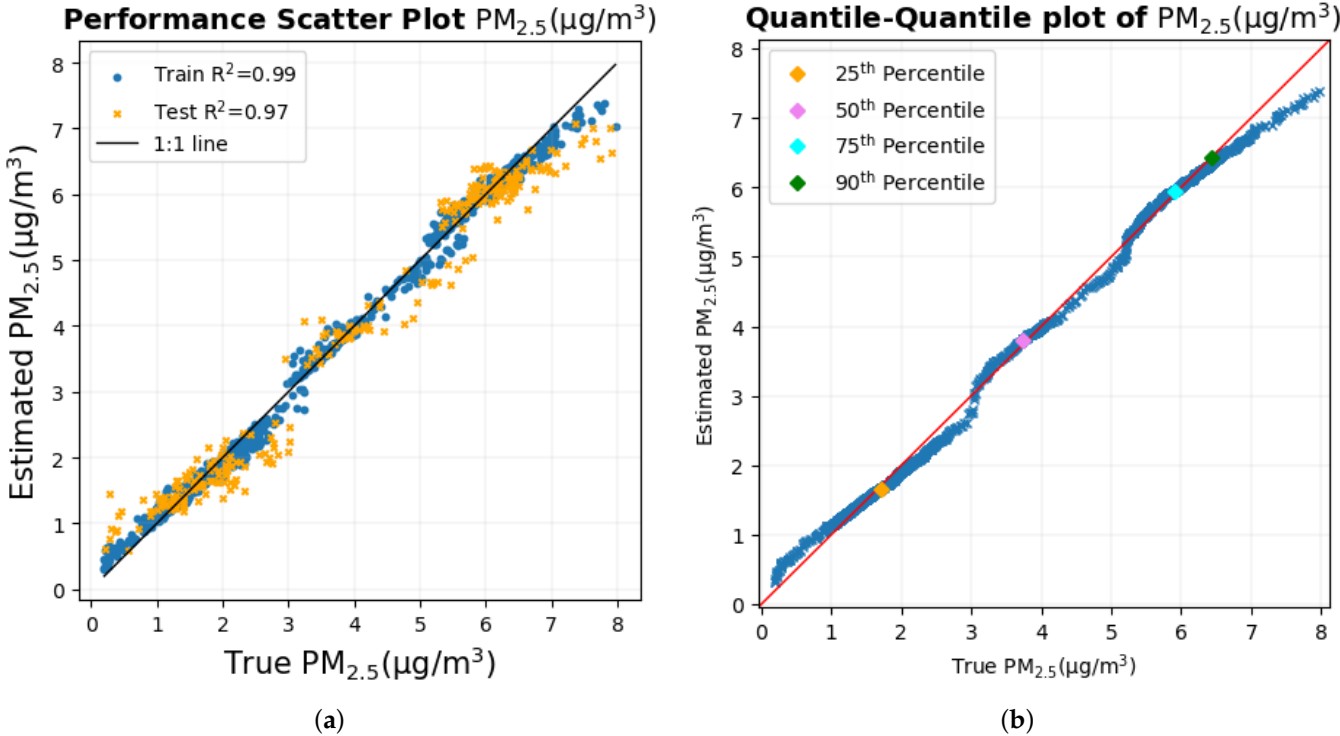

(**a**) (**b**)

**Figure 11.** (**a**) Scatter diagram of the true values of PM$_{2.5}$ against the estimated values of PM$_{2.5}$ with a black 1:1 line overlaid. (**b**) Quantile-quantile plot of the true values of PM$_{2.5}$ against the estimated values of PM$_{2.5}$ with a red 1:1 line overlaid.

A time series graph of the three gaseous pollutants is shown in Figure 12. The background of the time series plot is shaded with different colors, depending on different trials, and the trials have been separated with vertical black lines.

Figure 13 shows the time series plot of the true values of $PM_1$ and $PM_{2.5}$ overlaid with the estimated values of $PM_1$ and $PM_{2.5}$, respectively. Figures 12 and 13 show that the true values of the pollutant were close to the estimated values of the pollutant for most of the data set.

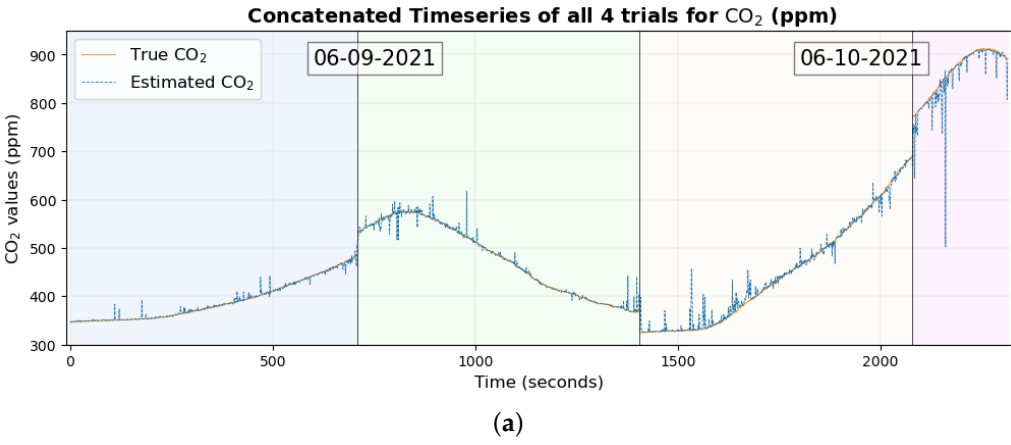

(**a**)

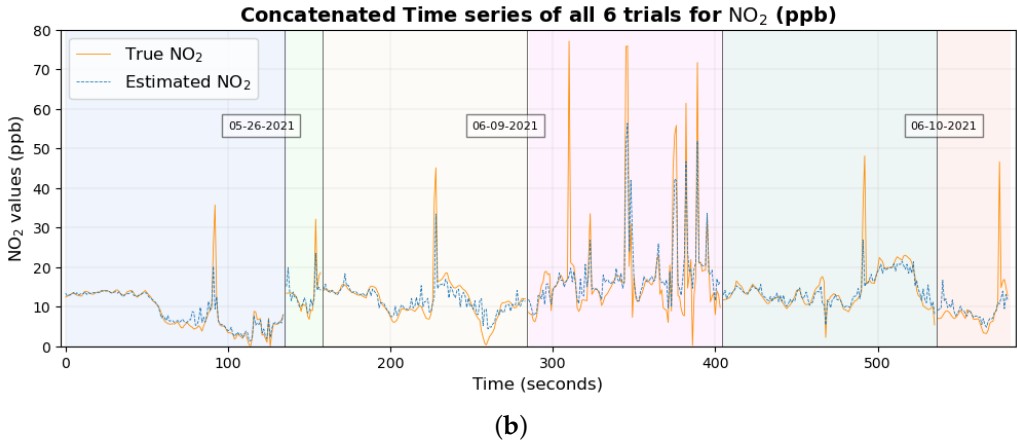

(**b**)

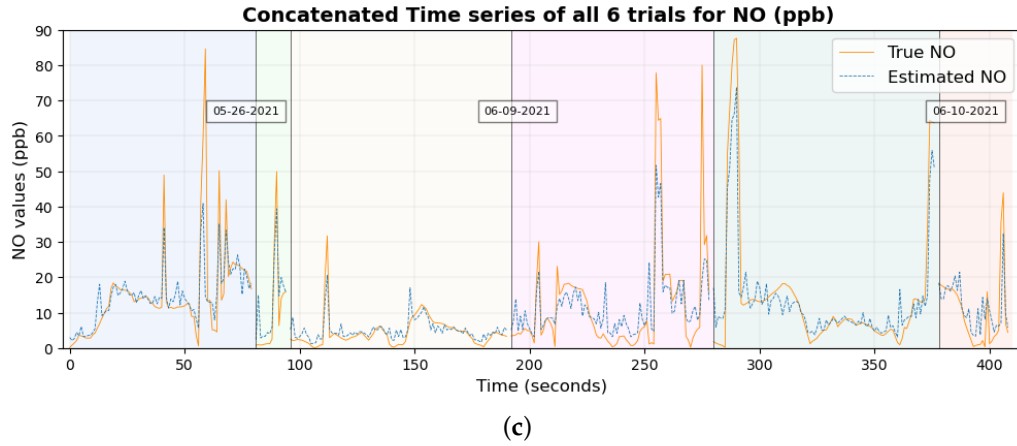

(**c**)

**Figure 12.** Time series plot of the true values of gaseous pollutants overlaid with estimated values of the pollutants for (**a**) $CO_2$, (**b**) $NO_2$, and (**c**) NO.

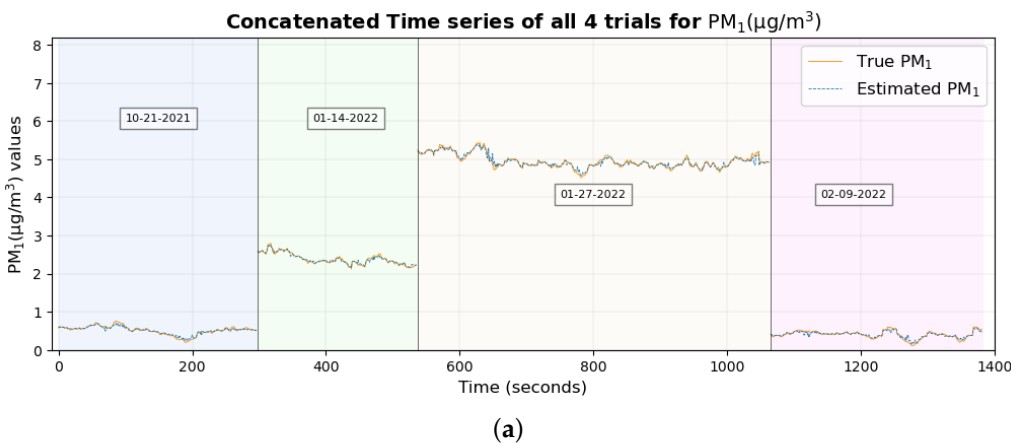

(**a**)

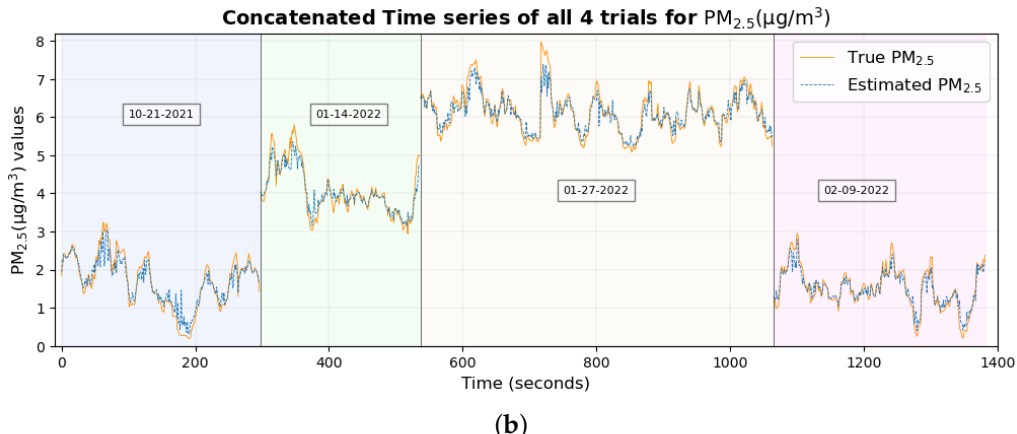

(**b**)

**Figure 13.** Time series plot of the true values of the pollutant overlaid with estimated values of the pollutant for (**a**) $PM_1$ and (**b**) $PM_{2.5}$.

*3.2. Using Easily Measurable Variables*

Now, let us focus on just the subset of biometric variables that can be easily measured using affordable sensors, for example, the respiration rate, $SpO_2$, heart rate, GSR, and skin temperature. All models using the reduced number of input features were trained with the same ensemble random forest regression algorithm from scikit-learn with optimized hyperparameters. Two of the hyperparameters that were optimized are *max_features* and *n_estimators*, and the best combination was found using *GridSearchCV*.

Table 8 shows the set of *n_estimators* and *max_features* provided to the *GridSearchCV* function and the best optimized parameters as well after the model was trained with 3-fold cross-validation.

Table 9 shows the results of $R^2$ and RMSE between the true values and the estimated values of the pollutant with the corresponding number of biometric variables used to estimate the pollutant.

The comparison of Table 9 with Table 5 shows that the test $R^2$ and RMSE for $PM_1$, $PM_{2.5}$, and $CO_2$ were very close to each other. Similar results can be seen for $NO_2$ and NO as well, for which the test $R^2$ was better, and the test RMSEs were very close to each other. The numbers for $NO_2$ and NO can change to some extent based on how the data are shuffled but with little disparity since the number of dimensions has now been significantly reduced. When the algorithm was run five times, the average $R^2$ between the actual and estimated values of $NO_2$ in the training and the test was 0.92 and 0.30, respectively, while that of NO was 0.95 and 0.55, respectively. Similarly, when the algorithm was run five times, the average RMSE between the true and estimated values of NO in the training and test set was 2.65 ppb and 7.81 ppb, while that of NO was 3.92 ppb and 9.23 ppb, respectively. The mentioned

values clearly indicate that the performance when the number of variables was reduced had increased for $NO_2$ and NO.

**Table 8.** Set of hyperparameters provided as inputs to *GridSearchCV* and the best possible parameter when considering reduced number of features.

| Pollutant | Set of *n_estimators* | Set of *max_features* | Folds for Cross-Validation | Total Number of Training | Optimized Parameter |
|---|---|---|---|---|---|
| $PM_1$ | 80, 90, 100, 110, 120, 150 | 2,3,4 | 3 | 54 | 120, 3 |
| $CO_2$ | 80, 90, 100, 110, 120, 150 | 2, 4, 5 | 3 | 54 | 100, 4 |
| $PM_{2.5}$ | 80, 90, 100, 110, 120, 150, 180 | 2, 3, 4 | 3 | 63 | 150, 3 |
| NO | 60, 70, 80, 90, 100, 110, 120 | 2, 4, 5 | 3 | 63 | 90, 4 |
| $NO_2$ | 80, 90, 100, 110, 120, 150 | 2, 3, 4 | 3 | 54 | 110, 2 |

**Table 9.** Quantification of the estimation of the pollutant using reduced number of variables with optimized hyperparameters of the random forest algorithm.

| Pollutant | Train $R^2$ | Test $R^2$ | Train RMSE | Test RMSE | Number of Biometrics Used |
|---|---|---|---|---|---|
| $PM_1$ | 0.99 | 0.99 | 0.03 µg/m$^3$ | 0.07 µg/m$^3$ | 4 |
| $CO_2$ | 0.99 | 0.98 | 7.62 ppm | 18.55 ppm | 5 |
| $PM_{2.5}$ | 0.99 | 0.96 | 0.16 µg/m$^3$ | 0.39 µg/m$^3$ | 4 |
| NO | 0.98 | 0.53 | 3.38 ppb | 12.70 ppb | 5 |
| $NO_2$ | 0.93 | 0.24 | 2.39 ppb | 10.38 ppb | 5 |

The biometric variables that have been considered for $CO_2$, $NO_2$, and NO now include the GSR, skin temperature, respiration rate, heart rate, and $SpO_2$, while those for $PM_1$ and $PM_{2.5}$ include skin temperature, heart rate, respiration rate, and $SpO_2$. A SHAP value beeswarm plot, scatter plot, and quantile–quantile plot of the gaseous pollutants estimated using the reduced number of variables is shown in Figure 14.

The orderings of the variables in the SHAP value were similar to each other for all gaseous pollutants. $SpO_2$ seems to be the lowest among all pollutants, and the elimination of this variable could have a small effect on the results. Since the number of dimensions has now been significantly reduced, the ordering will remain almost similar when the data are shuffled.

The scatter plot of $CO_2$ is similar to that when all variables were considered. As the $R^2$ value has increased for $NO_2$ and NO, the data points in the scatter plot are closer to the 1:1 line. Similarly, the structure of each quantile–quantile plot is similar for all gaseous pollutants when compared to the process in which all variables were considered.

Figure 15 shows the SHAP value beeswarm plot, scatter plot, and quantile–quantile plot when estimating $PM_1$ and $PM_{2.5}$ using only four biometric variables. The beeswarm plot in Figure 15a,d shows that skin temperature remains the main variable for estimating $PM_1$ and $PM_{2.5}$ with a very high SHAP value compared to the other variables. The overall structure of the scatter plot and the quantile–quantile plot of $PM_1$ and $PM_{2.5}$ also remains similar with a large portion of the data set close to the 1:1 black line and the 1:1 red line, respectively.

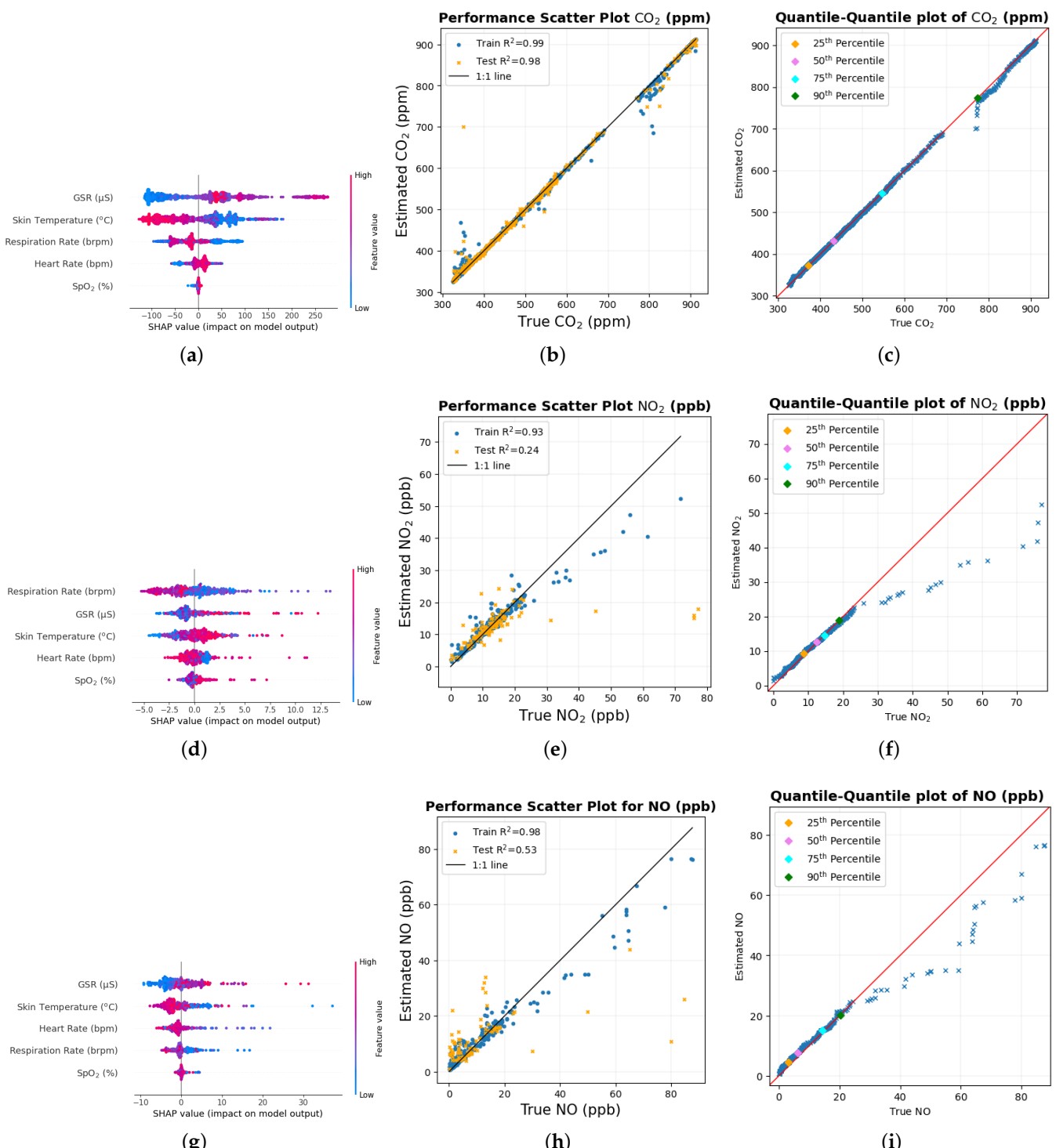

**Figure 14.** Top features and performance graphs using a reduced number of features: (**a**–**c**) to estimate inhaled $CO_2$, (**d**–**f**) to estimate inhaled $NO_2$, and (**g**–**i**) to estimate inhaled NO.

The time series plot with the reduced number of biometric variables to estimate $CO_2$, $NO_2$, and NO is shown in Figure 16. The time series plot with a reduced number of features to estimate $PM_1$ and $PM_{2.5}$ is shown in Figure 17.

Figure 16a shows that the difference between the true values and the estimated values of $CO_2$ is now smaller, as the true values and estimated values are much closer to each other compared to the time series when all features were considered. Similarly, the time series plot for $NO_2$ and NO is also similar to that when all variables were considered.

**Figure 15.** Top features and performance graphs using a reduced number of features: (**a**–**c**) to estimate inhaled $PM_1$ and (**d**–**f**) to estimate inhaled $PM_{2.5}$

The time series plot of $PM_1$ and $PM_{2.5}$ in Figure 17 shows that the true values and estimated values are close and similar to those when all features were considered.

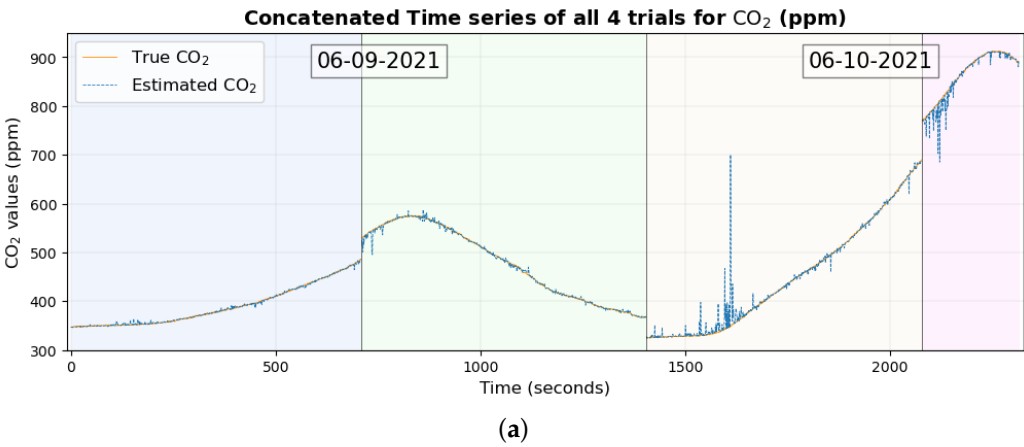

**Figure 16.** *Cont.*

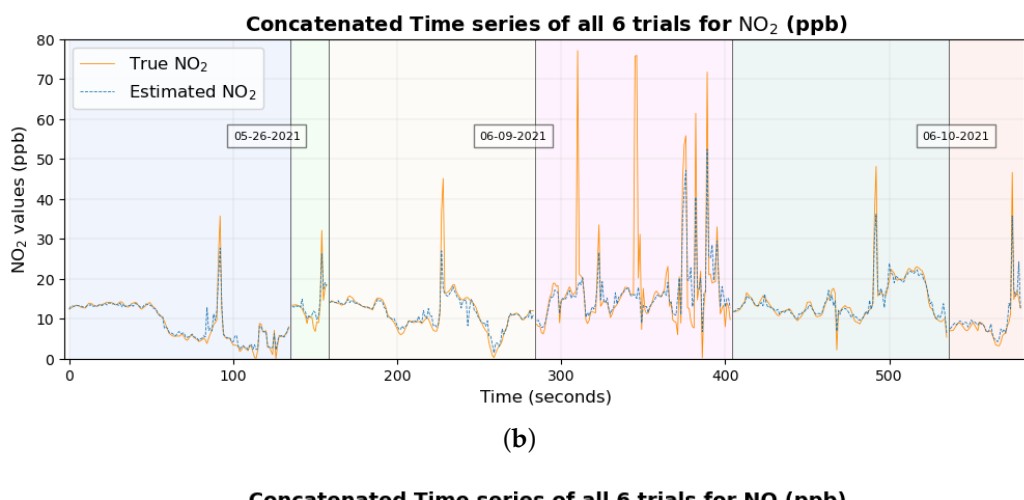

(**b**)

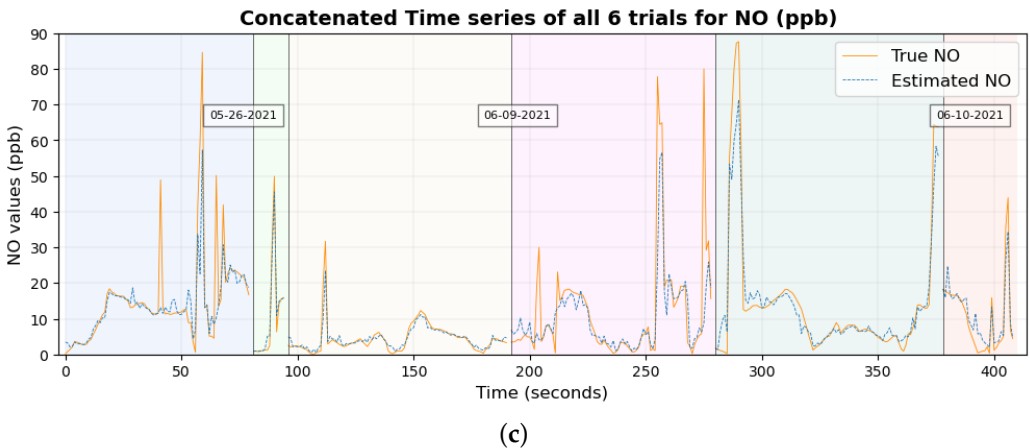

(**c**)

**Figure 16.** Time series plot of the true values of pollutants overlaid with the estimated values of pollutants using a reduced number of variables for (**a**) $CO_2$, (**b**) $NO_2$, and (**c**) NO.

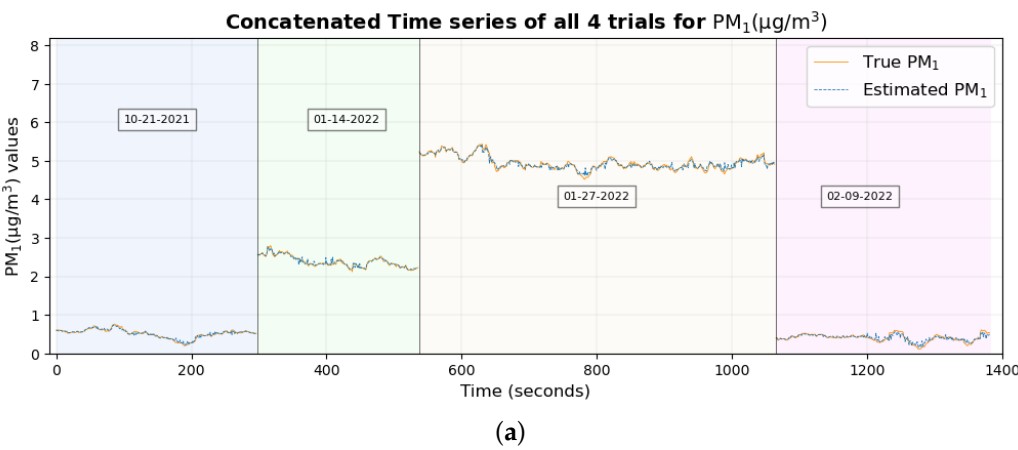

(**a**)

**Figure 17.** *Cont.*

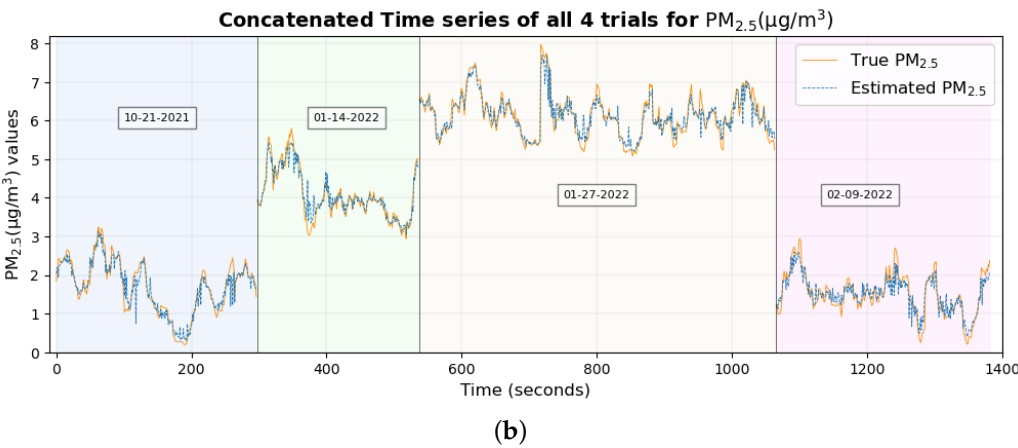

**(b)**

**Figure 17.** Time series plot of the true values of the pollutant overlaid with estimated values of the pollutant using a reduced number of variables for (**a**) $PM_1$ and (**b**) $PM_{2.5}$.

## 4. Discussion

The human body is a sensing system in itself, and it reacts to environmental variables and changes in them such as temperature, humidity, and air quality. It was previously shown that autonomous physiological and cognitive responses that result from the inhalation of particulate matter on a small temporal and spatial scale can be used to estimate $PM_1$ and $PM_{2.5}$ using machine learning models with very high accuracy [46] in a study that was limited to a single participant. The inclusion of multiple participants in the experimental static bike ride paradigm in which the measurement of $PM_1$ and $PM_{2.5}$ was performed shows that the methodology that was implemented on a single participant can be extended to multiple participants as well, producing even better results for $PM_1$ and $PM_{2.5}$ with an $R^2$ value of nearly 1 and a very low RMSE, as shown in Table 9. In fact, the results show that a few biometric variables are good enough to estimate $PM_1$ and $PM_{2.5}$ with similar results.

The time series plot of $PM_1$ and $PM_{2.5}$ in Figure 17a,b shows that their true values are very close to the estimated values for the majority of the data set without any significant differences, which explains their smallest RMSE among all pollutants. This supports the conclusion made previously [46] that two of the possible reasons why these estimates are highly accurate and precise could be (a) that these particulate matter are abundant and mix well with the ambient environment, thus having a higher probability of being inhaled by the participant and entering the sensors placed nearby and (b), with the minute size of $PM_{2.5}$, that these particulates, when inhaled, can reach deep into the lungs and bloodstream, creating many negative health effects [3,5,6], thereby impacting the human body to a large extent.

Air quality components include not only particulate matter but also gaseous pollutants such as $CO_2$, $NO_2$, and $NO$, which were included in this study. The methodology that was implemented to estimate and understand autonomous responses in the human body can be used for gaseous pollutants such as $CO_2$ as well. The $R^2$ value, which is nearly 1, between the true and estimated values of $CO_2$ in the test set using a small number of biometrics supports this claim, as shown in Table 9. Making the model simpler by considering a small number of biometrics also appears to have reduced the RMSE between the true and estimated values of $CO_2$, which can be seen clearly by comparing the time series in Figures 12a and 16a. Given the several physiological changes brought about by inhaling $CO_2$, such as changes in lung and cardiovascular function [33], cognitive issues [54], sweating [55], and the inflammation of airways, these autonomous responses can indeed be used to predict the concentration of $CO_2$ with high accuracy.

The results of estimating $NO_2$ and $NO$ for the entire range of data were not very accurate, as indicated by the value of $R^2$ and RMSE between the true and estimated values of the corresponding gas shown in Table 9. However, the scatter diagram of these two gases

in Figure 14e,h and the quantile–quantile plot of both of these gases in Figure 14f,i indicate that the prediction is reliable to some extent for lower values of the gas when there is a higher concentration of data, as the data points in the plot are close to their corresponding 1:1 line, respectively. As the number of data points decreases for higher values of these two gases, the data points in the scatter plot and the quantile–quantile plot deviate from their corresponding 1:1 line, with one possible reason being the very small number of data points for the machine learning model to learn from in this region of data. Moreover, Pearson's correlation coefficient (R) is highly susceptible to outliers when few data points deviate from the 1:1 line can largely affect the value of R. This could have possibly reduced the precision when the entire data set was considered for study. This claim is supported by the scatter diagram in Figure 14b and the quantile–quantile plot in Figure 14c of $CO_2$, where the data points deviate from the corresponding 1:1 line between 700 ppm and 800 ppm, one possible reason being the scarcity of data points in that region of data for the machine learning model to learn from and then to be tested on an independent test set. Improvements to the result in future work can possibly be made with either more expansive data collection or better machine learning models that can learn with a limited set of data to better minimize the error and then be tested in an independent test set.

Another possible reason for the results concerning NO and $NO_2$ not being highly accurate could also be that the autonomous responses during the process of data collection were dominated by $PM_1$ particles. As shown in the time series graph in Figure 16b,c, the concentration of $NO_2$ and NO was under 80 ppb and 90 ppb, respectively, with occasional high concentration. The concentration of $PM_1$ particles during these trials was between 0.708 and 7.655 $\mu g/m^3$. As mentioned before, since these minute particles, when inhaled, can pass through the nose and reach deep into the lungs and bloodstream, the immediate changes in the body were, thus, most likely dominated by these $PM_1$ particles for which the estimation of $PM_1$ using biometric variables was very high with the $R^2$ between the true and estimated values being 0.91 [46].

The result for all these air quality components shows that a small number of biometric variables used to estimate these pollutants provide similar and, in some cases, better results. In fact, the results are significantly better for NO and $NO_2$. Reducing the number of dimensions in a small data set, thus, seems to be more efficient in predicting the concentration, rather than a large number of input features. This aligns with Occam's razor principle suggesting that a simpler model usually generalizes well. Moreover, the reduction of the number of variables, that is, reducing the number of dimensions, was a necessity, given the small number of data sets compared to the large number of biometric variables for which data were collected.

There were a few limitations to this study that can possibly be removed in future work. One of them was the collection of data from a single participant for $CO_2$, $NO_2$, and NO. Multiple trials have been conducted to mitigate the issue. Future work can include more extensive data collection from multiple participants to provide further confirmation. However, the data collection in this experimental paradigm was performed on multiple days, with multiple trials under different environmental conditions; the results are, thus, likely to hold in a variety of environmental situations, probably except for situations with extreme weather. Moreover, due to the experimental paradigm involving a static bike ride, in which the study was conducted using multiple participants, and measurements of $PM_1$ and $PM_{2.5}$ were obtained, the results will also likely hold over a variety of populations. The other limitation of the study involved readings from some of the electrodes in the EEG headset that could be distorted due to activities such as blinking, head movement, swallowing, jaw clenching, neck movement, and tongue movement, which are frequent when a participant is cycling. This results in a lot of noise in the data that can be removed, but these activities are frequent, and the procedure can significantly reduce the number of data records. However, the results show that the removal of EEG data as biometric variables also yields similar results.

The methodology used in this study presents a unique application of machine learning. The use of biological measurements as input features for machine learning models can predict the concentration of air quality components such as $PM_1$, $PM_{2.5}$, and $CO_2$ with high degrees of accuracy. We can, thus, know the quality of air in microenvironments just by using a small set of biological measurements. Furthermore, with the use of predictor ranking, we can observe which biological parameter is most affected by these air quality components. Since the study was conducted outdoors, the participants were inhaling a mixture of varying pollutants. In order to study the direct effects of these pollutants, participants could be placed in a closed chamber with autonomous responses examined by artificially varying just one of the pollutants. A study can be conducted using just an EEG headset and observing how different areas of the brain can be affected when various components of air quality are inhaled. Future work can also conduct studies concerning other pollutants, such as lead, carbon monoxide, and volatile organic compounds.

Since this study was conducted on different days under different environmental conditions, confounding variables in the experimental setup were expected. For example, the ambient temperature can affect skin temperature and the GSR sensor as well. Future work can measure these environmental variables and identify these variables via causal analysis.

**Author Contributions:** Methodology, D.J.L., S.T. and T.L.; Software, S.R., S.T. and A.F.; formal analysis, S.R. and D.J.L.; data curation, S.T., D.J.L., L.O.H.W., A.F., T.L., M.L., J.S., A.A., J.W. and P.M.H.D.; writing—original draft preparation, S.R.; writing—review and editing, S.R., D.J.L., J.W. and L.O.H.W.; visualization, S.R.; supervision, D.J.L. All authors have read and agreed to the published version of the manuscript.

**Funding:** This research was funded by the following grants: the US Army (Dense Urban Environment Dosimetry for Actionable Information and Recording Exposure, U.S. Army Medical Research Acquisition Activity, BAA CDMRP Grant Log #BA170483), EPA 16th Annual P3 Awards Grant Number 83996501, entitled Machine Learning-Calculated Low-Cost Sensing, The Texas National Security Network Excellence Fund Award for Environmental Sensing Security Sentinels, and the SOFWERX Award for Machine Learning for Robotic Team and NSF Award OAC-2115094.

**Institutional Review Board Statement:** All experimental protocols were approved by the University of Texas at Dallas Institutional Review Board.

**Informed Consent Statement:** Informed consent was obtained from all the participants.

**Data Availability Statement:** The codes and data that were used to produce the results are publicly available at: https://github.com/mi3nts/Estimate-Inhaled-PM-and-Gases (accessed on 9 February 2024). The data set for $PM_1$ and NO is also available through Zenodo: https://zenodo.org/records/10639498 (accessed on 9 February 2024).

**Acknowledgments:** The authors acknowledge the OIT-Cyberinfrastructure Research Computing group at the University of Texas at Dallas and the TRECIS CC* Cyberteam (NSF 2019135) for providing HPC resources that were used in this study: https://utdallas.edu/oit/departments/circ/, (accessed on 9 February 2024).

**Conflicts of Interest:** The authors declare no conflicts of interest.

## Abbreviations

The following abbreviations are used in this manuscript:

| | |
|---|---|
| PM | Particulate matter |
| EEG | Electroencephalogram |
| ECG | Electrocardiogram |
| GSR | Galvanic skin response |
| $SpO_2$ | Blood oxygen saturation |
| RMSE | Root mean square error |

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
