# Peer review of "Quantifying Inhaled Concentrations of Particulate Matter, Carbon Dioxide, Nitrogen Dioxide, and Nitric Oxide Using Observed Biometric Responses with Machine Learning"

_biomedinformatics, doi:10.3390/biomedinformatics4020057_

Round 1

Reviewer 1 Report

Comments and Suggestions for Authors

Author Response

Thank you so much for taking the time to read our paper, it is much appreciated. We take your comments very seriously and have sought to address each of them. Thank you!

Reviewer 2 Report

Comments and Suggestions for Authors

In the paper “Quantifying Inhaled Concentrations of Particulate Matter, Carbon Dioxide, Nitrogen Dioxide, and Nitric Oxide Using Observed Biometric Responses with Machine Learning”, the authors used human biometrics to predict the concentrations of pollutants such as PM1, PM2.5, CO2, NO2 and NO in inhaled air. The paper demonstrated high prediction accuracy for many of the pollutants using the biometric data. There are a few comments from this reviewer.

1.     It’s interesting that biometric measurements can predict the pollutant concentrations, however, it’s unclear for the practical use of the prediction models.

2.     Since the study combined data across experiments, how batch effects were controlled for?

3.     For the results section, the detailed descriptions of the plots should be in the figure legends while the in-depth discussions for the pollutants and biometrics should be in the discussion section.

4. Detailed discussions are needed on the novelty and impacts of the study.

Author Response

(The authors gave the same response as above.)

Round 2

Reviewer 1 Report

Comments and Suggestions for Authors

The authors have addressed the concerns raised and improved the manuscript.

Reviewer 2 Report

Comments and Suggestions for Authors

All the comments from this reviewer has been addressed.